# Frequency-Limited Model Reduction for Linear Positive Systems: A Successive Optimization Method

**Yingying Ren** [1,2,*] **and Qian Wang** [1,3]

1    School of Automation and Electrical Engineering, University of Science and Technology Beijing, Beijing 100083, China
2    Shunde Innovation School, University of Science and Technology Beijing, Foshan 528399, China
3    Key Laboratory of Knowledge Automation for Industrial Processes of Ministry of Education, Beijing 100083, China
*    Correspondence: renyyc1994@163.com

**Abstract:** This paper studies frequency-limited model reduction for linear positive systems. Specifically, the objective is to develop a reduced-order model for a high-order positive system that preserves the positivity, while minimizing the approximation error within a given $H_\infty$ upper bound over a limited frequency interval. To characterize the finite-frequency $H_\infty$ specification and stability, we first present the analysis conditions in the form of bilinear matrix inequalities. By leveraging these conditions, we derive convex surrogate constraints by means of an inner-approximation strategy. Based on this, we construct a novel iterative algorithm for calculating and optimizing the reduced-order model. Finally, the effectiveness of the proposed model reduction method is illustrated with a numerical example.

**Keywords:** positive systems; frequency-limited model reduction; bilinear matrix inequalities; successive convex optimization algorithm

## 1. Introduction

In various practical fields, such as biology, chemistry, economics and engineering, it is often the case that certain quantities are potentially constrained to be non-negative. The population of a species in an ecosystem [1], the transmission speed that a signal travels in network communication [2], and the distribution of light energy radiated by a light source, to mention a few. This property arises from physical or biological constraints that preclude negative values; the systems that exhibit the positivity are commonly referred to as positive systems whose state variables and output trajectories take non-negative values, provided that the excitation inputs and initial states are non-negative [3–5]. The study of positive systems can be traced back to the early 1990s when Berman [6] introduced the concept of positive matrices and their applications to linear systems. Later, the theory of linear systems was further developed. It has become a vibrant research area in control theory and systems engineering, including modeling and analysis [7], control and optimization [8–11], and its applications [12]. In many cases, high-order models are frequently encountered to provide exact characterizations of dynamical behaviors, which poses a significant challenge for the analysis and synthesis of the concerned systems [13]. Therefore, it is crucial to simplify models of dynamical systems by exploring lower-order models that approximate the original high-order ones according to certain criteria [14,15]. Various model reduction methods, such as balanced truncation [16], Hankel-norm reduction [17], and moment matching [18] have been developed.

For a specific positive system, it is reasonably required to preserve the positivity when employing model-reduction methods. Moreover, it suffices to satisfy given approximation specifications within limited frequency ranges [19]; it is especially relevant in engineering applications where practical constraints limit the operating frequency range. There are

several model reduction approaches available to improve approximation performance over a limited frequency range, including methods such as frequency weighting [20] and frequency-specific balanced truncation [21]. Nevertheless, these methods cannot be directly applied to positive systems as they do not take the essential positivity of the reduced-order model into account. The generalized Kalman–Yakubovich–Popov (KYP) lemma is a powerful tool for characterizing the finite-frequency property, providing an equivalent condition for the solvability of frequency-limited specifications [22]. In the context of model reduction, this condition is expressed in a unified manner as a bilinear matrix inequality (BMI) involving coupled terms between two dependent decision variables [23]. To address this issue, some researchers have conservatively transformed the intractable BMI into a linear matrix inequality (LMI) by performing congruent transformation [24]. However, a potential drawback of this method is that the obtained LMI condition may always be infeasible. By alternately fixing one of the two coupled terms, D-K synthesis has been proposed to solve the BMI problems iteratively [25]; a major challenge of this approach lies in obtaining a feasible initial condition. Note that the introduction of additional positivity constraints and finite-frequency characterizations exacerbates the non-convexity, thereby making it difficult to explore a desired reduced-order model. Motivated by this, the paper aims to overcome this obstacle by developing a novel iterative procedure that can effectively solve the non-convex optimization problem.

In this paper, we aim to obtain a model with a lower order that preserves the positivity and captures the most significant behaviors within limited frequency intervals, which requires that the obtained model is positive and the resulting error system is stable with a given finite-frequency performance level. For this purpose, we establish conditions for the existence of such a reduced-order system in the form of BMIs. To address this problem, we propose a successive convex optimization (SCO) algorithm that iteratively solves a series of convex optimization sub-problems, each of which is an inner convex approximation of the original non-convex constraint. At each iteration, the solution to the previous step is used as a starting point for the next iteration until the convergence is reached. To summarize, this paper presents three contributions:

1.  A finite-frequency specification is employed to characterize the approximation error, improving the model reduction capability within limited frequency intervals.
2.  It is guaranteed that the reduced-order model maintains positivity, which retains the positive nature of the original system.
3.  An inner convexity strategy and associated SCO algorithm are proposed to obtain a desired reduced-order model without any parametrization techniques.

This paper is organized as follows: Section 2 introduces the problem statement and preliminary lemmas. Section 3 provides the conditions for the existence of a reduced-order model and a sequential algorithm for optimizing it. Section 4 gives an academic example to show the efficacy of the proposed method. Finally, Section 5 summarizes the contributions and discusses the limitations of the study and future work.

**Notations:** The symbols $\mathbb{R}$, $\mathbb{R}^n$, and $\mathbb{R}^{m \times n}$ represent the set of real numbers, $n$-dimensional column vectors, and $m \times n$-dimensional matrices, respectively. The notation $\mathbb{R}^n_+$ refers to a set of column vectors with element-wise positive entries. The symbols **I** and **0** denote the identical and zero matrices, respectively. For $M \in \mathbb{R}^{m \times m}$, $M > \mathbf{0}$ ($M < \mathbf{0}$) means that $M$ is positive-definite (negative-definite). He$\{M\} \triangleq M + M^T$, where $M^T$ is the transpose of $M$. Given a matrix $S \in \mathbb{R}^{m \times n}$, we use $S \succeq 0 (S \succ 0)$ to signify that all the elements are non-negative (positive).

## 2. Problem Statement and Fundamental Results

Consider a stable system ($\Sigma$) with a relatively high order:

$$\Sigma : \begin{cases} \lambda[x(t)] = Ax(t) + Bu(t), \\ y(t) = Cx(t) + Du(t), \end{cases} \tag{1}$$

where $x \in \mathbb{R}^{n_x}, u \in \mathbb{R}^{n_u}, y \in \mathbb{R}^{n_y}$ are the state, input and output vectors, respectively, and $[A, B; C, D]$ are the given constant matrices. The model $\Sigma$ provides a unified description for linear systems, that is, $\lambda[x(t)] \triangleq \dot{x}(t)$ for the continuous-time (CT) case and $\lambda[x(t)] \triangleq x(t+1)$ for the discrete-time (DT) case. Suppose that $\Sigma$ is a positive system whose definition is provided as follows:

**Lemma 1** ([26]). *The system $\Sigma$ is positive if all the state and output trajectories are non-negative, that is, $\forall t \geq 0, x(t) \in \mathbb{R}_+^{n_x}$ and $y(t) \in \mathbb{R}_+^{n_u}$, provided that $x(0) \in \mathbb{R}_+^{n_x}$ and $u(t) \in \mathbb{R}_+^{n_u}$.*

**Lemma 2** ([19]). *For the CT case, the system $\Sigma$ is positive if, and only if, $A$ is Metzler and $B \succeq 0$, $C \succeq 0, D \succeq 0$, whereas, for the DT case, the system $\Sigma$ is positive if, and only if, $A \succeq 0, B \succeq 0$, $C \succeq 0, D \succeq 0$.*

We intend to investigate an available model $\Sigma_r$ with a smaller order

$$\Sigma_r : \begin{cases} \lambda[x_r(t)] = A_r x_r(t) + B_r u(t), \\ y_r(t) = C_r x_r(t) + D_r u(t), \end{cases} \tag{2}$$

for approximating $\Sigma$ with a sufficiently small error, where $x_r \in \mathbb{R}^{n_r} (0 < n_r < n_x), y_r \in \mathbb{R}^{n_y}$ are reduced states and approximated outputs, respectively, and $[A_r, B_r; C_r, D_r]$ are unknown matrices. By defining $x_e(t) = \begin{bmatrix} x^T(t) & x_r^T(t) \end{bmatrix}^T$ and $e(t) \triangleq y(t) - y_r(t)$, the approximation error system can be expressed as

$$\Sigma_e : \begin{cases} \lambda[x_e(t)] = A_e x_e(t) + B_e u(t), \\ e(k) = C_e x_e(t) + D_e u(t), \end{cases} \tag{3}$$

where

$$A_e = \begin{bmatrix} A & \mathbf{0} \\ \mathbf{0} & A_r \end{bmatrix}, B_e = \begin{bmatrix} B \\ B_r \end{bmatrix}, C_e = \begin{bmatrix} C & -C_r \end{bmatrix}, D_e = D - D_r.$$

The transfer function of (3) is presented as:

$$\mathcal{G}_{ue}(\lambda) = C_e(\lambda \mathbf{I} - A_e)^{-1} B_e + D_e \tag{4}$$

where $\lambda$ denote the Laplace operator for the CT case and the $z$ operator for the DT case, respectively. For brevity, we give the following set to denote a positive system.

**Definition 1.** *The reduced-order system $\Sigma_r$ is positive if, and only if, $\begin{bmatrix} A_r & B_r \\ C_r & D_r \end{bmatrix} \in \mathbb{P}$, where*

$$\mathbb{P} \triangleq \begin{cases} \begin{bmatrix} A_r & B_r \\ C_r & D_r \end{bmatrix} : A_r \text{ is Metzler }, B_r \succeq 0, C_r \succeq 0, D_r \succeq 0 \\ \begin{bmatrix} A_r & B_r \\ C_r & D_r \end{bmatrix} : A_r \succeq 0, B_r \succeq 0, C_r \succeq 0, D_r \succeq 0 \end{cases} \tag{5}$$

There are two concerns when developing a reduced-order model $\Sigma_r$: First, it is expected that $\begin{bmatrix} A_r & B_r \\ C_r & D_r \end{bmatrix} \in \mathbb{P}$ to preserve the positivity of $\Sigma$. Second, in practical applications, it suffices to evaluate the approximation performance within a finite-frequency range. To provide an exact characterization for the property, we employ a frequency-limited $H_\infty$ specification

$$\|\mathcal{G}_{ue}(\lambda)\|_\infty^\Omega \triangleq \begin{cases} \sup_{\omega \in \Omega} \sigma_{max}[\mathcal{G}_{ue}(j\omega)] < \gamma, \text{(CT case)} \\ \sup_{\omega \in \Omega} \sigma_{max}[\mathcal{G}_{ue}(e^{j\omega})] < \gamma, \text{(DT case)} \end{cases} \tag{6}$$

throughout the paper, where $\Omega$ gives the frequency interval of interest, $\sigma_{max}$ denotes the maximum singular value, and $\gamma > 0$ is an index to be minimized. Therefore, we summarize the overall problem formulation as follows:

**Frequency-Limited Model Reduction:** Given a specific frequency range $\Omega$, explore a reduced-order model $\Sigma_r$ such that

1. The system $\Sigma_r$ is stable, and $\begin{bmatrix} A_r & B_r \\ C_r & D_r \end{bmatrix} \in \mathbb{P}$.

2. The approximation error system $\Sigma_e$ satisfies the finite-frequency criterion in (6).

Next, we introduce the generalized KYP lemma, which is essential to the analysis of finite-frequency specifications.

**Lemma 3** ([22]). *Given the state-space Equation (3) and a specific frequency range $\Omega$, the system $\Sigma_e$ has a guaranteed criterion $\|\mathcal{G}_{ue}(\lambda)\|_\infty^\Omega < \gamma$ if, and only if, there exist symmetric matrices $P, Q \in \mathbb{R}^{n_x + n_r}$ such that $Q > 0$ and*

$$\Psi^T \Phi \Psi < 0 \tag{7}$$

*where $\Psi = \begin{bmatrix} A_e^T & \mathbf{I} & C_e^T & \mathbf{0} \\ B_e^T & \mathbf{0} & D_e^T & \mathbf{I} \end{bmatrix}^T$, $\Phi = \text{diag}\{\Xi, \mathbf{I}, -\gamma^2 \mathbf{I}\}$, and $\Xi$ concerning different frequency ranges are listed in Table 1.*

**Table 1.** $\Xi$ with respect to different frequency ranges ($\omega_c = \frac{\omega_1 + \omega_2}{2}, \omega_a = \frac{\omega_2 - \omega_1}{2}$).

| Frequency Ranges | $\|\omega\| < \omega_l$ | $0 < \omega_1 < \omega < \omega_2 < \infty$ | $\|\omega\| > \omega_h$ |
|---|---|---|---|
| CT | $\begin{bmatrix} -Q & P \\ P & \omega_l^2 Q \end{bmatrix}$ | $\begin{bmatrix} -Q & P + j\omega_c Q \\ P - j\omega_c Q & -\omega_1 \omega_2 Q \end{bmatrix}$ | $\begin{bmatrix} Q & P \\ P & -\omega_h^2 Q \end{bmatrix}$ |
| DT | $\begin{bmatrix} P & Q \\ Q & -P - 2\cos \omega_l Q \end{bmatrix}$ | $\begin{bmatrix} P & e^{j\omega_c} Q \\ e^{-j\omega_c} Q & -P - 2\cos \omega_a Q \end{bmatrix}$ | $\begin{bmatrix} P & -Q \\ -Q & -P + 2\cos \omega_h Q \end{bmatrix}$ |

**Remark 1.** *Lemma 3 provides an exact characterization for (6), which differs from the bounded real lemma and has the potential to improve the approximation capability with less conservatism. Notice that the condition for the solvability of limited-frequency performance is expressed as a BMI condition, which is difficult to solve due to the non-convexity, and it becomes even more arduous when an extra constraint $G_r \in \mathbb{P}$ is imposed.*

## 3. Main Results

This section presents a method to design an appropriate reduced-order model. Firstly, an equivalent characterization for the system (3) is developed to parameterize the unknown matrices $A_r, B_r, C_r, D_r$. Then, the design conditions in the form of BMIs are established, ensuring that the reduced-order model maintains positivity and the error system has a given finite-frequency performance level. On this basis, we propose an iterative procedure to obtain a reduced-order model.

To simplify the problem, we denote $G_r = \begin{bmatrix} A_r & B_r \\ C_r & D_r \end{bmatrix}$ and

$$\bar{A} = \begin{bmatrix} A & \mathbf{0} \\ \mathbf{0} & \mathbf{0} \end{bmatrix}, \bar{B} = \begin{bmatrix} B \\ \mathbf{0} \end{bmatrix}, \bar{C} = \begin{bmatrix} C & \mathbf{0} \end{bmatrix}, \bar{D} = D,$$

$$\bar{F} = \begin{bmatrix} \mathbf{0} & \mathbf{0} \\ \mathbf{I} & \mathbf{0} \end{bmatrix}, \bar{H} = \begin{bmatrix} \mathbf{0} & -\mathbf{I} \end{bmatrix}, \bar{M} = \begin{bmatrix} \mathbf{0} & \mathbf{I} \\ \mathbf{0} & \mathbf{0} \end{bmatrix}, \bar{N} = \begin{bmatrix} \mathbf{0} \\ \mathbf{I} \end{bmatrix},$$

and, thus, (3) can be reformulated as $A_e = \bar{A} + \bar{F} G_r \bar{M}, B_e = \bar{B} + \bar{F} G_r \bar{N}, C_e = \bar{C} + \bar{H} G_r \bar{M}, D_e = \bar{D} + \bar{H} G_r \bar{N}$. According to Remark 1, the condition (7) cannot be directly solved for which the constrained $G_r \in \mathbb{P}$ is coupled with $P$ and $Q$. Therefore, we will provide feasible conditions in the following theorem.

**Theorem 1.** *Given a specified finite frequency range $\Omega$ and matrices $G_r^\kappa, P_s^\kappa, X^\kappa, Y^\kappa, Z^\kappa, U_s^\kappa, V_s^\kappa,$*
*$U^\kappa, V^\kappa$, a desired reduced-order model $G_r$ can be obtained if there exist matrices $P_s, X, Y, Z, P, Q, U_s,$*
*$V_s, U, V$ such that $P_s = P_s^T > 0, Q = Q^T > 0$ and*

$$G_r = \begin{bmatrix} A_r & B_r \\ C_r & D_r \end{bmatrix} \in \mathbb{P}, \tag{8}$$

$$\Gamma_s = \begin{bmatrix} \Gamma_{s1} & \Gamma_{s2} \\ \Gamma_{s2}^T & \Gamma_{s3} \end{bmatrix} < \mathbf{0} \tag{9}$$

$$\Gamma = \begin{bmatrix} \Gamma_1 & \Gamma_2 \\ \Gamma_2^T & \Gamma_3 \end{bmatrix} < \mathbf{0} \tag{10}$$

*where*

$$\Gamma_{s1} = \mathrm{He}\{P_s A_r^\kappa + P_s^\kappa(A_r - A_r^\kappa)\},$$

$$\Gamma_{s2} = \begin{bmatrix} \beta(P_s - P_s^\kappa) + (A_r - A_r^\kappa)^T (U_s^\kappa)^T & (A_r - A_r^\kappa)^T V_s^\kappa & \mathbf{0} \end{bmatrix},$$

$$\Gamma_{s3} = \begin{bmatrix} -\beta(U_s + U_s^T) & \mathbf{0} & U_s - U_s^\kappa \\ \star & -2V_s^\kappa + V_s & \mathbf{0} \\ \star & \star & -V_s \end{bmatrix},$$

$$\Gamma_1 = \begin{bmatrix} \Xi & \mathbf{0} & \mathbf{0} \\ \hdashline \mathbf{0} & C_e^T \\ \star & -\gamma^2\mathbf{I} & D_e^T \\ & D_e & -\mathbf{I} \end{bmatrix} + \mathrm{He}\left\{ \begin{bmatrix} -X & X(\bar{A} + \bar{F}G_r^\kappa\bar{M}) & X(\bar{B} + \bar{F}G_r^\kappa\bar{N}) & \mathbf{0} \\ -Y & Y(\bar{A} + \bar{F}G_r^\kappa\bar{M}) & Y(\bar{B} + \bar{F}G_r^\kappa\bar{N}) & \mathbf{0} \\ -Z & Z(\bar{A} + \bar{F}G_r^\kappa\bar{M}) & Z(\bar{B} + \bar{F}G_r^\kappa\bar{N}) & \mathbf{0} \\ \mathbf{0} & \mathbf{0} & \mathbf{0} & \mathbf{0} \end{bmatrix} \right\}$$

$$+ \mathrm{He}\left\{ \begin{bmatrix} (X^\kappa) \\ (Y^\kappa) \\ (Z^\kappa) \\ \mathbf{0} \end{bmatrix} \begin{bmatrix} \mathbf{0} & \bar{F}(G_r - G_r^\kappa)\bar{M} & \bar{F}(G_r - G_r^\kappa)\bar{N} & \mathbf{0} \end{bmatrix} \right\},$$

$$\Gamma_2 = \begin{bmatrix} \beta(X - X^\kappa) & \mathbf{0} & \mathbf{0} \\ \beta(Y - Y^\kappa) + \bar{M}^T(G_r - G_r^\kappa)\bar{F}^T(U^\kappa)^T & \bar{M}^T(G_r - G_r^\kappa)\bar{F}^T(V^\kappa)^T & \mathbf{0} \\ \beta(Z - Z^\kappa) + \bar{N}^T(G_r - G_r^\kappa)\bar{F}^T(U^\kappa)^T & \bar{N}^T(G_r - G_r^\kappa)\bar{F}^T(V^\kappa)^T & \mathbf{0} \\ \mathbf{0} & \mathbf{0} & \mathbf{0} \end{bmatrix},$$

$$\Gamma_3 = \begin{bmatrix} -\beta(U + U^T) & \mathbf{0} & U - U^\kappa \\ \star & -2V^\kappa + V & \mathbf{0} \\ \star & \star & -V \end{bmatrix}$$

**Proof of Theorem 1.** By Lyapunov stability theory, the asymptotic stability of the reduced-order system (2) can be ensured if, and only if, there exists a Lyapunov matrix $P_s = P_s^T$ satisfying

$$A_r^T P_s + P_s A_r < \mathbf{0} \tag{11}$$

and (7) can be further expressed as

$$\begin{bmatrix} A_e & B_e \\ \mathbf{I} & \mathbf{0} \\ \mathbf{0} & \mathbf{I} \end{bmatrix}^T \underbrace{\left( \mathrm{diag}\{\Xi, \mathbf{0}\} + \mathrm{diag}\left\{\mathbf{0}, \begin{bmatrix} C_e & D_e \\ \mathbf{0} & \mathbf{I} \end{bmatrix}^T \Pi \begin{bmatrix} C_e & D_e \\ \mathbf{0} & \mathbf{I} \end{bmatrix}\right\} \right)}_{\Upsilon} \begin{bmatrix} A_e & B_e \\ \mathbf{I} & \mathbf{0} \\ \mathbf{0} & \mathbf{I} \end{bmatrix} < \mathbf{0} \tag{12}$$

Let $\mathcal{H} = \begin{bmatrix} -\mathbf{I} & A_e & B_e \end{bmatrix}^T$, and its null space denoted as $\mathcal{H}^\perp = \begin{bmatrix} A_e & B_e \\ \mathbf{I} & \mathbf{0} \\ \mathbf{0} & \mathbf{I} \end{bmatrix}^T$. Thus, (14) is

equivalent to

$$\mathcal{H}^\perp \Gamma (\mathcal{H}^\perp)^T < \mathbf{0} \tag{13}$$

By Finsler's lemma, we have

$$\mathcal{H}^{\perp}\Gamma(\mathcal{H}^{\perp})^T < \mathbf{0} \Leftrightarrow \Gamma + \mathcal{H}\mathcal{X} + \mathcal{X}^T\mathcal{H}^T < \mathbf{0} \tag{14}$$

where $\mathcal{X} = \begin{bmatrix} X^T & Y^T & Z^T \end{bmatrix}$. It follows that (11) and (14) can be presented in a unified form of

$$\mathbb{L}_J + \text{He}\{\mathbb{A}_J N_J \mathbb{B}_J\} < \mathbf{0}, J = 1, 2. \tag{15}$$

where

$$\mathbb{L}_1 = \mathbf{0}, \mathbb{A}_1 = P_s, \mathbb{B}_1 = A_r, N_1 = \mathbf{I},$$

$$\mathbb{L}_2 = \begin{bmatrix} -Q & P & \mathbf{0} & \mathbf{0} \\ \star & \omega_l^2 Q & \mathbf{0} & C_e^T \\ \star & \star & -\gamma^2 \mathbf{I} & D_e^T \\ \star & \star & \star & -\mathbf{I} \end{bmatrix} + \text{He}\left\{ \begin{bmatrix} -X & X\bar{A} & X\bar{B} & \mathbf{0} \\ -Y & Y\bar{A} & Y\bar{B} & \mathbf{0} \\ -Z & Z\bar{A} & Z\bar{B} & \mathbf{0} \\ \mathbf{0} & \mathbf{0} & \mathbf{0} & \mathbf{0} \end{bmatrix} \right\},$$

$$\mathbb{A}_2 = \begin{bmatrix} X & Y & Z & \mathbf{0} \end{bmatrix}^T, \mathbb{B}_2 = \begin{bmatrix} \mathbf{0} & \bar{F}G_r\bar{M} & \bar{F}G_r\bar{N} & \mathbf{0} \end{bmatrix}, N_2 = \mathbf{I}.$$

It is observed that $\mathbb{A}_l$ is coupled with $\mathbb{B}_l$. For given feasible matrices $P_s^\kappa, G_r^\kappa, X^\kappa, Y^\kappa, Z^\kappa$, we can reformulate (15) as

$$\mathbb{L} + \text{He}\{\mathbb{A}N\mathbb{B}^\kappa + \mathbb{A}^\kappa N(\mathbb{B} - \mathbb{B}^\kappa)\} + \text{He}\{(\mathbb{A} - \mathbb{A}^\kappa)N(\mathbb{B} - \mathbb{B}^\kappa)\} < \mathbf{0}. \tag{16}$$

We notice that $\mathbb{L} + \text{He}\{\mathbb{A}N\mathbb{B}^\kappa + \mathbb{A}^\kappa N(\mathbb{B} - \mathbb{B}^\kappa)\}$ is linear, whereas $\text{He}\{(\mathbb{A} - \mathbb{A}^\kappa)N(\mathbb{B} - \mathbb{B}^\kappa)\}$ is bilinear. According to [23], the bilinear term can be further decomposed into

$$\text{He}\{(\mathbb{A} - \mathbb{A}^\kappa)N(\mathbb{B} - \mathbb{B}^\kappa)\}$$
$$= \begin{bmatrix} (\mathbb{A} - \mathbb{A}^\kappa)^T \\ (\mathbb{B} - \mathbb{B}^\kappa) \end{bmatrix}^T \begin{bmatrix} \mathbf{0} & N \\ N^T & \mathbf{0} \end{bmatrix} \begin{bmatrix} (\mathbb{A} - \mathbb{A}^\kappa)^T \\ (\mathbb{B} - \mathbb{B}^\kappa) \end{bmatrix}$$
$$= \begin{bmatrix} (\mathbb{A} - \mathbb{A}^\kappa)^T \\ (\mathbb{B} - \mathbb{B}^\kappa) \end{bmatrix}^T \begin{bmatrix} \beta N & \beta N \\ U^T & -U \end{bmatrix} \begin{bmatrix} \frac{1}{\beta}(U + U^T)^{-1} & \mathbf{0} \\ \mathbf{0} & -\frac{1}{\beta}(U + U^T)^{-1} \end{bmatrix} \begin{bmatrix} \beta N & \beta N \\ U^T & -U \end{bmatrix}^T \begin{bmatrix} (\mathbb{A} - \mathbb{A}^\kappa)^T \\ (\mathbb{B} - \mathbb{B}^\kappa) \end{bmatrix} \tag{17}$$
$$\leq \frac{1}{\beta}\left(\beta(\mathbb{A} - \mathbb{A}^\kappa)N + (\mathbb{B} - \mathbb{B}^\kappa)^T U^T\right)\left(U + U^T\right)^{-1}\left(\beta(\mathbb{A} - \mathbb{A}^\kappa)N + (\mathbb{B} - \mathbb{B}^\kappa)^T U^T\right)^T$$

where $U$ serves as an auxiliary variable and $U + U^T > \mathbf{0}$, without loss of generality. Defining $\Delta\mathbb{A}^\kappa \triangleq \mathbb{A} - \mathbb{A}^\kappa, \Delta\mathbb{B}^\kappa \triangleq \mathbb{B} - \mathbb{B}^\kappa$, and applying the above approximation strategy, results in

$$\begin{bmatrix} \mathbb{L} + \text{He}\{\mathbb{A}N\mathbb{B}^\kappa + \mathbb{A}^\kappa N\Delta\mathbb{B}^\kappa\} & \beta\Delta\mathbb{A}^\kappa N + (U\Delta\mathbb{B}^\kappa)^T \\ \star & -\beta(U + U^T) \end{bmatrix} < \mathbf{0}. \tag{18}$$

Similarly, we decompose $U$ as $U = U^\kappa + \Delta U^\kappa$, and thus

$$\begin{bmatrix} \mathbb{L} + \text{He}\{\mathbb{A}N\mathbb{B}^\kappa + \mathbb{A}^\kappa N\Delta\mathbb{B}^\kappa\} & \beta\Delta\mathbb{A}^\kappa N + (U^\kappa\Delta\mathbb{B}^\kappa)^T \\ \star & -\beta(U + U^T) \end{bmatrix} + \text{He}\left\{ \begin{bmatrix} (\Delta\mathbb{B}^\kappa)^T \\ \mathbf{0} \end{bmatrix} \begin{bmatrix} \mathbf{0} & (\Delta U^\kappa)^T \end{bmatrix} \right\} < \mathbf{0}. \tag{19}$$

According to $\text{He}\{\mathbb{U}\mathbb{V}\} \leq \mathbb{U}\mathbb{M}\mathbb{U}^T + \mathbb{V}^T\mathbb{M}^{-1}\mathbb{V}$, one obtains that

$$\begin{bmatrix} \mathbb{L} + \text{He}\{\mathbb{A}N\mathbb{B}^\kappa + \mathbb{A}^\kappa N\Delta\mathbb{B}^\kappa\} & \beta\Delta\mathbb{A}^\kappa N + (U^\kappa\Delta\mathbb{B}^\kappa)^T \\ \star & -\beta(U + U^T) \end{bmatrix}$$
$$+ \begin{bmatrix} (\Delta\mathbb{B}^\kappa)^T \\ \mathbf{0} \end{bmatrix} V \begin{bmatrix} (\Delta\mathbb{B}^\kappa) & \mathbf{0} \end{bmatrix} + \begin{bmatrix} \mathbf{0} \\ (\Delta U^\kappa) \end{bmatrix} V^{-1} \begin{bmatrix} \mathbf{0} & (\Delta U^\kappa)^T \end{bmatrix} < \mathbf{0}. \tag{20}$$

which is equivalent to

$$\begin{bmatrix} \mathbb{L} + \text{He}\{\mathbb{A}N\mathbb{B}^\kappa + \mathbb{A}^\kappa N\Delta\mathbb{B}^\kappa\} & \beta\Delta\mathbb{A}^\kappa N + (U^\kappa\Delta\mathbb{B}^\kappa)^T & (\Delta\mathbb{B}^\kappa)^T & \mathbf{0} \\ \star & -\beta(U + U^T) & \mathbf{0} & (\Delta U^\kappa) \\ \star & \star & -V^{-1} & \mathbf{0} \\ \star & \star & \star & -V \end{bmatrix} < \mathbf{0} \tag{21}$$

We observe that $-V^{-1}$ is convex, implying $-V^{-1} \leq -2(V^\kappa)^{-1} + (V^\kappa)^{-1} V (V^\kappa)^{-1}$ for a given $V^\kappa$. Replacing $-V^{-1}$ with $-2V^\kappa + (V^\kappa)^{-1} V (V^\kappa)^{-1}$ and performing the congruent transformation $\mathrm{diag}\{\mathbf{I}, \mathbf{I}, V^\kappa, \mathbf{I}\}$ on (21), we can obtain (9) and (10). □

In Theorem 1, we first establish sufficient and necessary conditions (11) and (14) for the existence of a desired model $\Sigma_r$ in the form of BMIs. Given feasible solutions $P_s^\kappa, G_r^\kappa$, $X^\kappa, Y^\kappa, Z^\kappa$ satisfying $\mathbb{L}_s + \mathrm{He}\{\mathbb{A}_s^\kappa N_s \mathbb{B}_s^\kappa\} < \mathbf{0}, \mathbb{L} + \mathrm{He}\{\mathbb{A}^\kappa N \mathbb{B}^\kappa\} < \mathbf{0}$, the BMI conditions can be rewritten as the sum of linear and residual terms. By virtue of the convex-approximation strategy, we propose an SCO algorithm that iteratively solves a series of convex optimization sub-problems,

$$\begin{cases} \min \gamma \\ \text{s.t. } P_s = P_s^T > \mathbf{0}, Q = Q^T > \mathbf{0}, (8)\text{–}(10). \end{cases} \tag{22}$$

each of which is an inner convex approximation of the original non-convex constraint. At each iteration, the solution to the previous step is used as a starting point for the next iteration until the convergence is reached. To be concise, the above procedures can be summarized in the following algorithm.

**Remark 2.** *As pointed out in Algorithm 1, it is crucial to have an admissible $G_r^0$ for optimizing the subsequent solutions. However, one can see that obtaining such an initial condition is equally as intricate as solving the original problem. It can be shown that the objective function decreases with each iteration of the SCO algorithm, and according to the convergence property, we can obtain an initial solution by simply choosing a stable model $G_r^0 \in \mathbb{P}$. This significantly eases the challenge of implementing the proposed algorithm.*

---

**Algorithm 1** SCO algorithm for calculating reduced-order models

---

**Require:** $\epsilon$: tolerable bound; *IMax*: maximal iteration limit

1: Given a feasible solution $G_r^0$, solve the convex optimization problem

$$\begin{cases} \min \gamma \\ \text{s.t. } (A_s^0)^T P_s + P_s A_s^0 < \mathbf{0}, \\ \mathrm{diag}\{\Xi, -\gamma^2 \mathbf{I}, \mathbf{I}\} + \mathrm{He}\left\{ \begin{bmatrix} X & \mathbf{0} \\ Y & \mathbf{0} \\ Z & \mathbf{0} \\ \mathbf{0} & \mathbf{I} \end{bmatrix} \begin{bmatrix} -\mathbf{I} & (\bar{A} + \bar{F}G_r^0\bar{M}) & (\bar{B} + \bar{F}G_r^0\bar{N}) & \mathbf{0} \\ \mathbf{0} & (\bar{C} + \bar{H}G_r^0\bar{M}) & (\bar{D} + \bar{H}G_r^0\bar{N}) & -\mathbf{I} \end{bmatrix} \right\} < \mathbf{0} \end{cases} \tag{23}$$

to obtain the initial condition $P_s^0, X^0, Y^0, Z^0$. Fix $\kappa \leftarrow 1$

2: **while** $\kappa < IMax$ **do**

3:     Solve (22) to the solution $(P_s, G_r, X, Y, Z)$ and optimum $\gamma$ at $\kappa + 1$-iteration.

4:     **if** $\frac{\gamma^\kappa - \gamma^{\kappa+1}}{\gamma^\kappa} \leq \epsilon$ **then**

5:         $\gamma^* \leftarrow \gamma^{\kappa+1}, G_r^* \leftarrow G_r^{\kappa+1}$

6:         **break;**

7:     **end if**

8:     $\kappa \leftarrow \kappa + 1$

9: **end while**

10: **Result** $G_r^*$: the desired $\Sigma_r$; $\gamma^*$: optimal finite-frequency performance index.

---

## 4. Simulation

In the simulation part, a system $\Sigma$ describing the compartmental network with two sub-systems is considered, and its state-space realization is given as follows:

$$
A = \begin{bmatrix}
-1.5 & 0.6 & 1.0 & 0 & 0 & 0 \\
0.3 & -1.9 & 0.2 & 0 & 0 & 0 \\
0.2 & 0.5 & -2.7 & 1 & 0 & 0 \\
0 & 0 & 0.5 & -3 & 0.6 & 0.5 \\
0 & 0 & 0 & 0.4 & -1.6 & 0.3 \\
0 & 0 & 0 & 0.6 & 0.5 & -1.6
\end{bmatrix}, B = \begin{bmatrix}
1 & 0 \\
0 & 1 \\
0 & 0 \\
0 & 0 \\
0 & 0 \\
0 & 0
\end{bmatrix}, \tag{24}
$$

$$
C = \begin{bmatrix} \mathbf{I}_{2\times2} & \mathbf{0}_{2\times4} \end{bmatrix}, D = \mathbf{0}_{2\times2}.
$$

The example is drawn from [19]. The restricted frequency is taken as $[0,2]\mathrm{rad/s}$. To develop a reduced-order model for approximating (24), we first adopt the initial condition as

$$
A_r^0 = \begin{bmatrix} -1.9301 & 0.7093 \\ 0.7188 & -1.1613 \end{bmatrix}, B_r^0 = \begin{bmatrix} 0.0294 & 0.0562 \\ 0.0847 & 0.0379 \end{bmatrix},
$$
$$
C_r^0 = \begin{bmatrix} 1.8925 & 3.7318 \\ 2.6822 & 1.2840 \end{bmatrix}, D_r^0 = \begin{bmatrix} 0.2669 & 0.0002 \\ 0.0231 & 0.2462 \end{bmatrix}. \tag{25}
$$

Using the initial model and applying Algorithm 1, we optimize the reduced-order model as

$$
A_r = \begin{bmatrix} -1.7544 & 0.3616 \\ 0.5570 & -1.2572 \end{bmatrix}, B_r = \begin{bmatrix} 0.0004 & 0.1864 \\ 0.1458 & 0.0001 \end{bmatrix},
$$
$$
C_r = \begin{bmatrix} 0.1540 & 5.9325 \\ 4.8997 & 0.0754 \end{bmatrix}, D_r = \begin{bmatrix} 0.0333 & 0.0001 \\ 0.0042 & 0.0226 \end{bmatrix}. \tag{26}
$$

with the optimal $\gamma^* = 0.0207$. By virtue of Lemma 2, it can be verified that the obtained reduced-order model (26) is positive. To validate the convergence property of the proposed algorithm, Figure 1 depicts the evolution of $\gamma$ by configuring the maximum number of iterations $IMax = 140$ and ignoring the tolerable bound $\epsilon$. As shown in Figure 1, the proposed algorithm exhibits a monotonically decreasing trend in $\gamma$ and converges to a fixed point eventually. Moreover, the result presents a significant improvement and provides a more exact approximation over the initial reduced-order model.

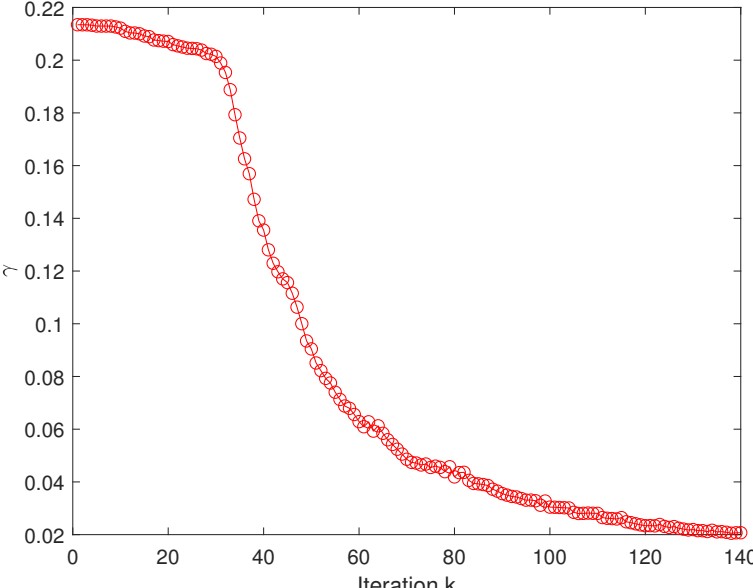

**Figure 1.** Evolution $\gamma$ by the proposed algorithm.

To illustrate the accuracy of (26) approximating (24), we provide Figures 2 and 3 that demonstrate the singular value curves of $\mathcal{G}_{uy}(\lambda)$ with $\mathcal{G}_{uy_r}(\lambda)$ and $\mathcal{G}_{ue}(\lambda)$, respectively. It can be observed that, in the given frequency range ($[0, 2]$ rad/s shaded region), the singular value curves of the reduced-order system $\Sigma_r$ closely match those of the original system $\Sigma$, while the actual maximum singular values of the error system $\Sigma_e$ are strictly upper-bounded by $\gamma^* = 0.0207$. Based on the above analysis, one can conclude that the proposed reduced-order model is effective in approximating a high-order system with significant small errors.

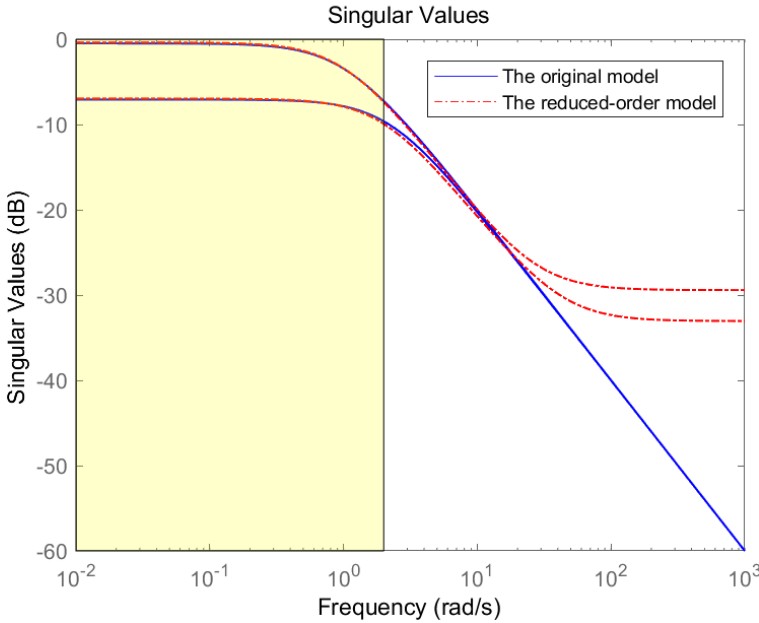

**Figure 2.** Singular value curves of the original and reduced-order systems.

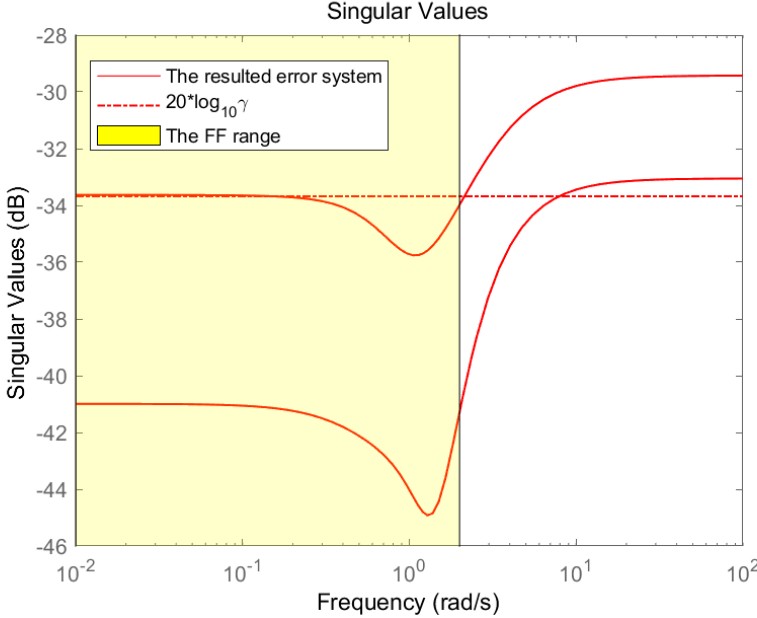

**Figure 3.** Singular value curves of the associated error system.

## 5. Conclusions

This paper has considered the positivity-preserving model reduction for positive systems within limited frequency regions. To be specific, the positivity of the reduced-order model has been formulated as an element-wise positivity constraint, while the finite-frequency specification for the error system has been translated into BMI conditions. To provide a unified framework for addressing the BMI problems, we have proposed an SCO algorithm that serves as a promising solution for developing reduced-order models with positivity constraints and finite-frequency characterizations. Finally, the usefulness of the proposed method has been validated through a compartmental network example.

**Author Contributions:** Methodology, Y.R.; validation, Q.W.; writing—original draft preparation, Y.R.; writing—review and editing, Y.R. All authors have read and agreed to the published version of the manuscript.

**Funding:** This research was funded by the National Natural Science Foundation of China under Grant 62103041, and by the Postdoctoral Research Foundation of Shunde Innovation School, University of Science and Technology Beijing under Grant 2021BH010.

**Institutional Review Board Statement:** Not applicable.

**Informed Consent Statement:** Not applicable.

**Data Availability Statement:** Not applicable.

**Conflicts of Interest:** The authors declare no conflict of interest.

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
