# Peer review of "Frequency-Limited Model Reduction for Linear Positive Systems: A Successive Optimization Method"

_applsci, doi:10.3390/app13064039_

Round 1
Reviewer 1 Report
In this paper, the authors studied frequency-limited model reduction for positive linear systems. The manuscript is poorly organized, with unclear data, confusing results and poor-quality figures. More importantly, I see very few scientific results and little, if any, novelty. I suggest rejecting this manuscript.
Reviewer 2 Report
This paper investigates a method to obtain a reduced-order model for linear positive systems within a frequency range.
The methodology defines the unknown matrices of the reduced model with frequency-limited h_inf specification as Bilinear Matrix Inequalities (BMIs). As a result, BMIs establish sufficient and necessary conditions for the existence of a reduced positive model.
The BMIs are decomposed and rewritten as the sum of linear and residual terms. Then, the authors propose a successive convex optimization algorithm to solve the equivalent BMIs to obtain the reduced-order model.
The overall paper is concise and the methodology describes in detail.
I don't have many comments or suggestions:
- Possibly more recent and relevant references on reduced-order models show that the topic is active.
- The introduction may be more descriptive as regards the pros and cons of survey methods.
Reviewer 3 Report
the paper is good , just improve the introduction about the history of the topic in the other researches.
Round 2
Reviewer 1 Report
I thank the authors for clarifying certain points. I suggest that this manuscript be accepted in its present form.